# Lung Aeration in COVID-19 Pneumonia by Ultrasonography and Computed Tomography

**DOI:** 10.3390/jcm11102718

**Published:** 2022-05-11

**Authors:** Alexandros Kalkanis, Christophe Schepers, Zafeiris Louvaris, Laurent Godinas, Els Wauters, Dries Testelmans, Natalie Lorent, Pierre Van Mol, Joost Wauters, Walter De Wever, Christophe Dooms

**Affiliations:** 1Department of Respiratory Diseases, University Hospitals, Katholieke Universiteit Leuven, 3000 Leuven, Belgium; laurent.godinas@uzleuven.be (L.G.); els.wauters@uzleuven.be (E.W.); dries.testelmans@uzleuven.be (D.T.); natalie.lorent@uzleuven.be (N.L.); pierre.vanmol@uzleuven.be (P.V.M.); christophe.dooms@uzleuven.be (C.D.); 2Department of Radiology, University Hospitals, Katholieke Universiteit Leuven, 3000 Leuven, Belgium; christophe.schepers@uzleuven.be (C.S.); walter.dewever@uzleuven.be (W.D.W.); 3Department of Rehabilitation Sciences, Faculty of Movement and Rehabilitation Sciences, Research Group for Rehabilitation in Internal Disorders, Katholieke Universiteit Leuven, 3000 Leuven, Belgium; zafeiris.louvaris@kuleuven.be; 4Laboratory of Translational Genetics, VIB—KU Leuven Center for Cancer Biology, 3000 Leuven, Belgium; 5Department of Internal Medicine, University Hospitals, Katholieke Universiteit Leuven, 3000 Leuven, Belgium; joost.wauters@uzleuven.be

**Keywords:** lung ultrasound, COVID-19 pneumonia, imaging, radiology, computed tomography

## Abstract

We conducted a prospective single-center observational study to determine lung ultrasound reliability in assessing global lung aeration in 38 hospitalized patients with non-critical COVID-19. On admission, fixed chest CT scans using visual (CTv) and software-based (CTs) analyses along with lung ultrasound imaging protocols and scoring systems were applied. The primary endpoint was the correlation between global chest CTs score and global lung ultrasound score. The secondary endpoint was the association between radiographic features and clinical disease classification or laboratory indices of inflammation. Bland–Altman analysis between chest CT scores obtained visually (CTv) or using software (CTs) indicated that only 1 of the 38 paired measures was outside the 95% limits of agreement (−4 to +4 score). Global lung ultrasound score was highly and positively correlated with global software-based CTs score (r = 0.74, CI = 0.55–0.86; *p* < 0.0001). Significantly higher median CTs score (*p* = 0.01) and lung ultrasound score (*p* = 0.02) were found in severe compared to moderate COVID-19. Furthermore, we identified significantly lower (*p* < 0.05) lung ultrasound and CTs scores in those patients with a more severe clinical condition manifested by SpO_2_ < 92% and C-reactive protein > 58 mg/L. We concluded that lung ultrasound is a reliable bedside clinical tool to assess global lung aeration in hospitalized non-critical care patients with COVID-19 pneumonia.

## 1. Introduction

The histopathology of early severe acute respiratory syndrome coronavirus 2 (SARS-CoV-2) interstitial pneumonia is characterized by an exudative inflammation consisting of an accumulation of monocytes in the alveolar cavities and by monocytes and lymphocytes centered on small blood vessels infiltrating widened alveolar septa [1,2]. The common radiographic manifestation of pneumonia in the setting of COVID-19 is the appearance of parenchymal opacities in the periphery of the lungs. Chest X-ray is often insensitive in early phase parenchymal lung disease [3]. Low-dose chest computed tomography (CT) is considered in the diagnostic algorithm of hospitalized symptomatic patients with COVID-19 infection and pneumonia [4,5], even though it is not always required for the initial diagnosis or treatment of non-critical COVID-19 infection. CT images acquired early in the course of pneumonia typically reveal a unilateral focal ground-glass opacity, quickly evolving into bilateral multilobar ground-glass opacities with peripheral or posterior distribution. As the disease progresses, ground-glass opacities may progress to consolidative pulmonary opacities and mixed patterns [6,7,8].

CT scanning is not available in all emergency departments. Therefore, alternative imaging modalities to identify and manage these infections are highly desirable. Indeed, lung ultrasound (LUS) can provide information on interstitial-alveolar syndrome, lung consolidation and pleural effusion, and has an established value in the evaluation of pulmonary edema, ARDS, and pneumonia [9,10]. According to the International Consensus Conference on lung ultrasound, B-lines are discrete laser-like vertical hyperechoic artefacts that appear from the pleural line and move synchronously with lung sliding [9]. 

Along these lines, lung interstitial syndrome is defined by an increased number of B-lines indicating the accumulation of extravascular fluid or inflammatory cells in the pulmonary interstitial space or alveoli and having a ground-glass opacity appearance with or without thickening of the interlobular septa on a chest CT scan. LUS is more accurate than standard chest X-ray or physical examination, and nearly as accurate as chest CT scan to detect community-acquired pneumonia [10,11,12,13], with the main disadvantage being its lower sensitivity in detecting deeper consolidations. The bedside utility of LUS has, therefore, consistently been suggested during the COVID-19 pandemic [14]. A prospective description of LUS findings in COVID-19 is not yet available. In addition, a LUS score can give a quantitative global assessment of lung aeration: an increase in LUS score indicates a decrease in lung parenchymal aeration. 

Accordingly, this study aimed to prospectively compare LUS with CT to detect loss of lung aeration in patients with COVID-19 pneumonia with lung opacity on chest CT scan as the gold standard, the latter computed with specific software (CTs) and validated with the visual method (CTv). It was reasoned that the LUS score would be associated with CTs scan measurements for evaluating lung aeration to assist the management of patients with COVID-19.

## 2. Materials and Methods

### 2.1. Study Design

This was a prospective single-center observational study on non-critical hospitalized patients admitted to a COVID-19 ward between 2 April 2020 and 24 April 2020. The study is part of a larger observational study that was registered at ClinicalTrials.gov, number NCT04327570. Data in Table 1 (baseline demographics, clinical and laboratory findings) as well as the LUS scores of the 38 patients of this study appeared in a recent publication of our group, which aimed to explore the LUS scores for the rapid assessment of the severity of SARS-CoV-2 pulmonary infection in patients hospitalized with COVID-19 pneumonia [15]. Furthermore, chest computed tomography data have not appeared anywhere in that, or in any other, report. Eligible patients for lung ultrasound evaluation met the following inclusion criteria: SARS-CoV-2 infection confirmed by a positive RT-PCR for SARS-CoV-2 RNA of a nasopharyngeal swab; low dose chest CT scan performed on admission to the emergency room; low dose chest CT scan performed within 24 h from the LUS; and presence of CT image abnormalities consistent with viral pneumonia. Exclusion criteria were: pulmonary edema; >24 h interval since chest CT scan; CT findings known to the sonographer; and patients admitted to the non-critical COVID-19 ward for reason of palliative care or sedation. This study was approved by the Institutional Review Board of the University Hospitals KU Leuven (study ID s60207). All participants provided written informed consent. 

### 2.2. Demographic, Clinical, and Laboratory Data

Patient demographic, clinical, and laboratory data were collected on the day of emergency room admission and collected from the electronic medical file of the patients. Clinical classification was applied to summarize patients’ condition at the time of hospital admission, according to the American Thoracic Society (ATS) ≥ 3 minor criteria for defining severe community-acquired pneumonia, and the China National Health Commission (NHC) clinical case classification considering severe COVID-19 cases who met at least one of the following conditions: (1) respiratory rate ≥ 30/min, or (2) oxygen saturation (resting state) ≤ 93%, or (3) PaO2/FiO2 ≤ 300 mmHg [16,17].

### 2.3. Chest Computed Tomography and Score Assessment

All CT scans were performed using a Siemens SOMATOM Definition Flash, dedicated to the COVID-19 emergency department of our institution. This was part of the emergency department (ED) planning of our tertiary university hospital, in order to immediately isolate patients with flu-like symptoms while all the needed tests were performed. All patients underwent a low-dose non-contrast CT of the chest in the inspiration phase with the following scan protocol: slice thickness and increment: 1 mm/0.7 mm (lung and mediastinal window), pitch: 1.2, collimation: 128 × 0.6 mm, rotation time: 0.5 s. The dose protocol (kV and mAs) was: <50 kg: 80 kV and 30 mAs; between 50 and 80 kg: 120 kV and 20 mAs; >80 kg: 140 kV and 28 mAs.

Chest radiologists independently and blinded to the lung ultrasound findings performed qualitative and quantitative evaluations of lung parenchyma opacities on the CT scan. Opacities of interest were low-density ground-glass opacity (GGO) and high-density consolidations. GGO was defined as hazy increased lung attenuation with preservation of bronchial and vascular margins, whereas consolidation was defined as an increase in parenchymal opacification with obscuration of margins of vessels and airway walls. The Syngo. VIA CT Pneumonia Analysis software program prototype was used to measure the percentage of lung parenchyma opacity [18]. Based on 3D segmentations of lungs, lobes, and pneumonia lesions, an artificial intelligence algorithm of the CT Pneumonia Analysis program automatically identified and quantified increased attenuation areas of the lung parenchyma (GGO and consolidations) on axial CT data with slice thicknesses up to 5 mm, and quantified, lobe-wise, the extent of these lung parenchyma opacities.

A CT score was assigned by converting the estimated (visual or CTv) and the measured (software or CTs) percentage of lung parenchyma opacity for each lobe into a 5-point scale: a score of 0 for 0% lung opacity, 1 for 1% to <5% lung opacity, 2 for 5% to 25% lung opacity, 3 for 26 to 50% lung opacity, 4 for 51 to 75% lung opacity, and 5 for 76 to 100% lung opacity [19]. The total CT score is the sum of the individual lobar scores and can range from 0 (no area with an increase in lung opacity) to 25 (all five lobes show more than a 75% increase in lung opacity) (see Figure A1, Figure A2, Figure A3, Figure A4 and Figure A5). 

### 2.4. Lung Ultrasonography and Score Assessment

Lung ultrasound was performed using a GE Healthcare LOGIQ E9 ultrasound system, dedicated to exclusive use at the non-critical COVID-19 wards of our institution with all unnecessary parts removed, and a curved 3.5-MHz array probe. All LUS examinations were performed within 12 h of the initial CT scan. To correctly identify the artifactual images of the lungs, the harmonic imaging was removed, and the reject post-processing was lowered. The focus was set at the level of the pleural line and depth was set at 15 cm from the pleural line. 

All lung ultrasound examinations were performed bedside in full personal protection equipment (PPE) and scored by one physician who remained blinded to the chest CT images. A 12-region lung ultrasound scanning method was used [20,21]. Each systematically examined hemithorax consisted of six regions: anatomical landmarks set by anterior and posterior axillary lines defined anterior, lateral, and posterior regions, which were each divided into superior and inferior. Patients were examined in supine and lateral position; the latter to examine the posterior regions. All intercostal spaces in all 12 regions were explored via both longitudinal and cross-sectional views, to perform a comprehensive examination [22]. 

In each of the 12 regions and during an entire respiratory cycle the most pathologic out of 4 ultrasound lung aeration patterns was considered representative for the entire region and classified as a score of 0 for normal aeration (lung sliding with A-lines); a score of 1 for moderate loss of aeration (≥3 well-spaced B-lines, or B1); a score of 2 for severe loss of aeration (coalescent B-lines including white lung, or B2); and a score of 3 for consolidation (hyperechoic lung tissue). A LUS score ranging from 0 to 36 (LUS) was calculated as the sum of each of the 12 regions. In addition, pleural fluid was registered if present. Representative ultrasound images from each of the 12 regions were extracted from the machine and stored in the electronic medical file of each patient.

### 2.5. Statistical Analysis

A Shapiro–Wilk test was applied to test the normality of the data. This analysis identified that CT (visually and software-based) and LUS scores were normally distributed. We did not perform a sensitivity analysis on the different sample sizes required for different levels of correlation between LUS and CT scores to determine the optimal sample size of this study to detect statistical significance. Nevertheless, the sample size calculation was based on the objective to detect at least a correlation coefficient of 0.5 for LUS and CT scores based on a previous study from our group, which indicated significant associations between LUS score and clinical outcomes (correlation r ranged between 0.48 and 0.58) in patients hospitalized for COVID-19 pneumonia [15]. A minimum required sample size for this study was 37 for a power of 90% and alpha level of significance of 0.05 [23]. Given the possibility of 5% dropouts, a sample size of 40 patients were recruited to address the aim of the study. Quantitative variables are summarized as mean (and SD) or median (and interquartile range, IQR 25–75%) for Gaussian or skewed distribution, respectively. Comparisons were performed with the Mann–Whitney test for skewed distributions. All tests were two-sided and statistical significance was determined as *p*-value < 0.05. Correlations between LUS and CT scores were evaluated by the Pearson’s correlation coefficient in case of two quantitative normally distributed variables. The Bland–Altman analysis was utilized to measure the agreement in aeration assessment between the two CT scoring methods, and 95% limits of agreement were calculated as the mean difference (1.96 × SD). A satisfactory agreement between CT scores measured visually (CTv) and CT scores calculated by the software (CTs) was considered when the difference between CTv and CTs measurements did not significantly vary from zero. For this purpose, one-sample *t*-test among the difference between CTv and CTs measurements and zero value was performed. All statistical analyses were performed with a statistical software package, GraphPad Prism version 5.0 for Mac, GraphPad Software, San Diego, CA, USA.

## 3. Results

### 3.1. Patient Characteristics

Forty consecutive patients were eligible and consented. Two subjects were excluded as they were not meeting the study entry criteria and 38 subjects were analyzed. Demographic, clinical, and laboratory data for the 38 study participants are presented in Table 1. In >90% of hospitalized subjects fever and/or respiratory symptoms (cough, dyspnea) were the presenting symptoms, while in the remaining subjects atypical symptoms, such as loss of appetite or confusion, were attributed to COVID-19 pneumonia. The WHO clinical disease state score, the China NHC clinical case classification, and the ATS community-acquired pneumonia severity index demonstrated a mild disease state with score of four in 79% of subjects, a severe clinical case in 76% of subjects, and a severe pneumonia in 18% of subjects, respectively. All subjects were admitted to the COVID-19 ward for disease monitoring.

### 3.2. Chest Computed Tomography (CT) and Lung Ultrasound (LUS) Scores 

Chest CT and lung ultrasound findings are depicted in Table 2. The median time interval between low-dose CT scan and LUS was 20 h (IQR 16–22). In all subjects the lung parenchymal opacities on the chest CT scan were located at least in the outer part of the hemithorax. Major descriptive radiographic findings included ground-glass opacity on the chest CT scan in 36 (95%) of subjects, and B-lines on LUS in 37 (97%) of subjects (Figure 1 and Figure 2). While GGO on the chest CT scan was present in 95% of subjects, this was the predominant abnormal CT finding in 84%. Similarly, LUS observed a B1 or B2 pattern in at least one of the twelve regions in 92% and 82% of subjects, respectively. A predominant presence out of 12 regions for the B1 or B2 pattern was observed in 61% and 42% of subjects, respectively, as in some subjects both patterns were equally predominantly present (e.g., 5 out of 12 regions for both B1 and B2 pattern). Consolidation on the chest CT scan was observed in 29% and the predominant abnormal finding in 16% of subjects. Similarly, LUS observed consolidation in 29% and was the predominant abnormal finding in 11% of subjects. A more global assessment of lung aeration was provided by quantification with a scoring system. Mean (±SD) loss of aeration score was 6.6 (±2.8) out of 25 points for CT_V_, 6.6 (±3.2) out of 25 points for CT_S_, and 11 (±5.3) out of 36 points for LUS. 

### 3.3. Agreement between CT-Software (CTs) and CT-Estimated (CTV) Scores

The CTv score was strongly correlated with the CTs score (r = 0.76, *p* < 0.0001; Figure 3). The mean difference between CT-software (CTs) and CT-estimated (CTv) score was zero points, indicating no average systematic measurement error or bias; the difference between the two methods did not vary statistically significantly from zero (*p* = 1.0, Figure 4). The size of the measurement error or 95% limits of agreement from −4 to +4 points was rather wide, implicating disagreement between the two scoring methods. The software-based method was used as the gold standard to compare with LUS. 

### 3.4. Correlation between Lung Ultrasound Score (LUS) and CT-Software (CT_S_) Score and Other Clinical Variables 

The CT_S_ score was highly and positively correlated with the global LUS score (r = 0.74, CI = 0.55–0.86, *p* < 0.0001; Figure 5). 

The association between the radiographic assessment of loss of lung aeration and baseline demographics, clinical classifications, or laboratory findings of inflammation is depicted in Table 3. No association was found between a radiographic quantitative evaluation and demographic findings of age, sex, or BMI. The median CTs and LUS score in China NHC severe-type clinical COVID-19 cases was 7 (IQR 5–10) and 11 (IQR 8–15), respectively, which was significantly higher (*p* < 0.05) than that of moderate severity cases (four with IQR 3–6, and nine with IQR 3–10, respectively). Furthermore, we identified significantly lower LUS and CTs scores in those patients with worsened clinical condition manifested by SpO2 < 92% and CRP > 58 mg/L (Table 3). No significant differences were found in LUS and CTs scores in patients with lower, as compared to patients with higher, neutrophil-to-lymphocyte ratio or D-dimers, probably related to a clinical context of hospitalized patients excluding critical-type intensive care unit COVID-19 cases. 

## 4. Discussion

Our study demonstrated satisfactory agreement between CTs and CTv in the assessment of lung aeration in patients with COVID-19; hence, the software-based method (CTs) was used as the gold standard to compare with LUS. We found a strong correlation for loss of lung aeration between a quantitative chest CTs score and a global LUS score obtained by a sonographer blinded to the chest CT scan.

Two types of disagreement between the two scoring systems used should be acknowledged. First, different scoring mechanisms and scales were developed and applied. The LUS scoring system assigns a score to a certain region (not anatomical lobe) considering the worst finding for rating, independently of its dimension [22]. Second, the depth of inspection for LUS is limited to the outer part of the hemithorax, while a CT scan evaluates the entire hemithorax [13]. This may lead to an overestimation of loss of aeration with higher LUS scores, certainly for the non-critical patients with a severe disease stage that is often limited to the outer hemithorax. Despite these flaws, our clinical findings support a global LUS score as a reliable bedside clinical assessment tool in hospitalized non-critical patients with COVID-19. Baseline higher than median (>92%) SpO2 and lower than median blood CRP value was significantly associated with lower radiographic (both CTs and LUS) scores of loss of lung aeration. The median CTs score and LUS score in severe-type was significantly higher than moderate COVID-19, a finding that was not observed when a non-COVID-19 ATS community-acquired pneumonia severity classification was applied.

The adoption of lung ultrasound at a point-of-care setting in a COVID-19 internal medicine ward has been proposed as the stethoscope of the 21st century to visualize the global lung aeration in SARS-CoV-2 pneumonia and assess changes or resolution of lung opacities over time [14,24]. Our findings contribute to the potential application of LUS as a bedside clinical tool for longitudinal monitoring of SARS-CoV-2 pneumonia, which should be evaluated in further prospective studies. 

Different point-of-care lung ultrasound scanning protocols have been described for qualitative evaluation of the lung. The BLUE-protocol decision tree is performed on acute dyspneic patients who will be admitted to the ICU. A systematic six-regions lung ultrasound examination is performed for immediate diagnosis of the main causes of acute respiratory failure [25]. A systematic 12-regions lung ultrasound examination has been prospectively evaluated in an intensive care unit setting for early diagnosis and monitoring of ventilator-associated pneumonia, and for the determination of lung aeration changes during spontaneous breathing weaning from mechanical ventilation [26,27,28]. A lung ultrasound scoring system has been described for this 12-region method to quantify the assessment of the lung [21,22,28]. The calculation of a LUS score allows semi-quantification of the global assessment of lung aeration regardless of etiology: an increase in LUS score indicates a decrease in lung aeration. Inter-observer agreements between physicians for 12-region LUS analysis and scoring was kappa 0.77 to 0.84 in blinded prospective research [29,30]. We decided to use the 12-region lung ultrasound examination with global LUS score. Patients hospitalized in a non-critical COVID-19 ward are likely in a stable clinical condition and able to turn into the lateral position used for the posterior lung surface examination. This extended range of examination is essential as COVID-19 is often characterized by bilateral multilobar opacities with a peripheral and/or posterior distribution. This LUS method examines the entire lung surface and gives a detailed image of aeriation loss, making it suitable for a qualitative and quantitative correlation with chest CT scan in a COVID-19 population. 

The strengths of this study are the prospective design with fixed imaging protocols including detailed evaluations with scoring system, and a LUS operator blinded to the chest CT images. Our study also has limitations. We acknowledge that our data are preliminary and larger studies are necessary to confirm the role of lung ultrasound in the management of COVID-19. Nevertheless, our data support the previous literature and further indicates the use of bedside ultrasound for the early diagnosis in patients who presented to the emergency department with COVID-19 pneumonia [24,31,32]. Moreover, this is a single-center study, and one expert sonographer performed all image acquisitions, the latter in order to minimize the exposure of health-care professionals and use of PPE. This can also justify the necessary time delay between the performance of the CT and LUS in our study. Larger studies are needed to support the significant associations between LUS and clinical outcomes we found in our study and further promote the use of LUS as prognostication in terms of severity of COVID-19 pneumonia and its trajectory. In our study, we do not conduct repeat imaging at different time points during the hospital stay to examine whether improvements in GGO were also correlated with improvements in LUS. Additionally, we cannot generalize our findings to more severe patients admitted directly to critical care. Finally, the sensitivity of LUS is the highest for the diagnosis of normally aerated tissue or pleural effusion, while false negative findings can be seen for alveolar-interstitial and consolidated tissue not reaching the pleural borders or when an affected lung area is surrounded by alveolar gas.

## 5. Conclusions

In conclusion, we found that the lung ultrasound score correlated strongly with the chest CT scan for the evaluation of COVID-19 with the added advantage of ease of use at point-of-care and the absence of radiation exposure. LUS is a reliable bedside clinical tool to evaluate global lung aeration and might be suitable as an alternative imaging modality for COVID-19 lung disease monitoring.

## 6. Patents

No patents resulting from the work are reported in this manuscript.

## Figures and Tables

**Figure 1 jcm-11-02718-f001:**
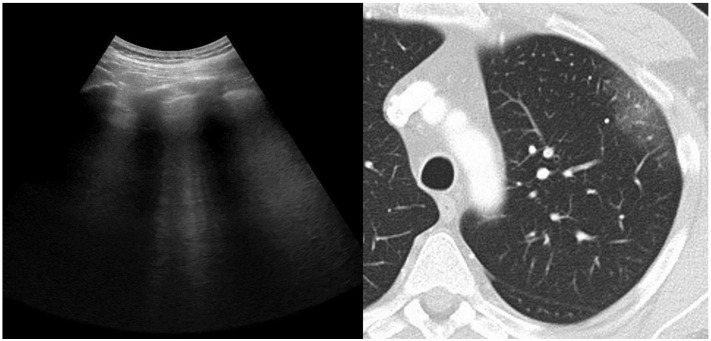
**Left panel**: LUS demonstrating thickening of the pleural line and intercostal predominantly well-spaced B-lines or B1 pattern. **Right panel**: CT scan demonstrating bilateral pure GGO.

**Figure 2 jcm-11-02718-f002:**
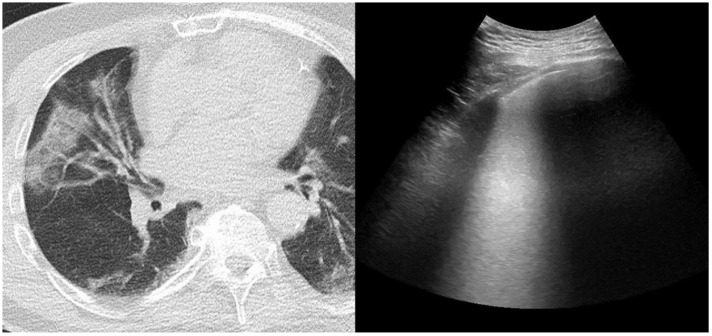
Left panel: CT scan demonstrating bilateral GGO with tendency of consolidation. **Right panel**: LUS demonstrating thickening of the pleural line and intercostal predominantly coalescent B-lines or B2 pattern.

**Figure 3 jcm-11-02718-f003:**
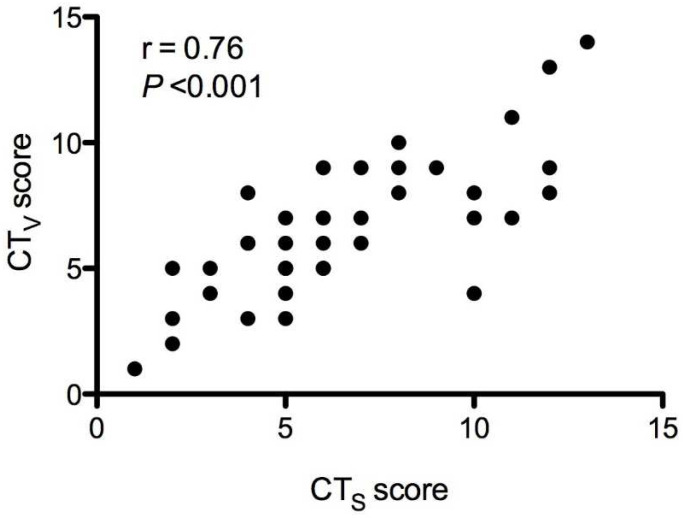
Pearson’s correlation for CTs and CTv scores.

**Figure 4 jcm-11-02718-f004:**
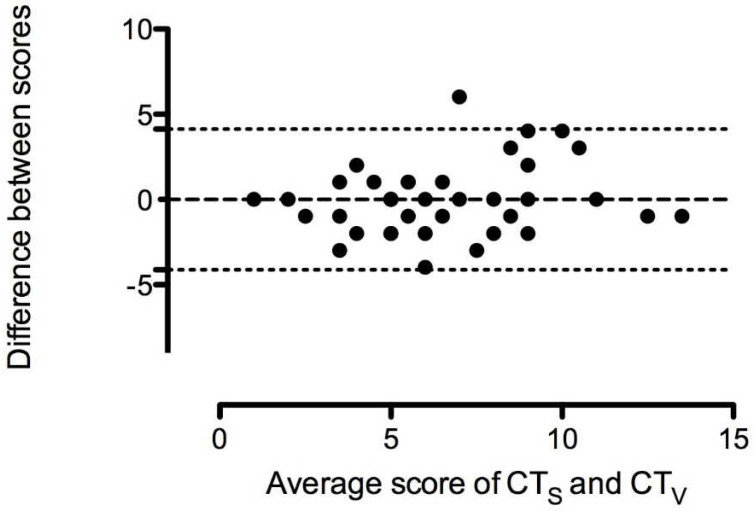
Bland–Altman plot of difference (CTs–CTv) against mean for CTs lung opacity score versus CTv lung opacity score. The dashed line represents the mean difference (bias 0.000), and the dotted lines represent the 95% limits of agreement (−4 to +4) between the paired measurements.

**Figure 5 jcm-11-02718-f005:**
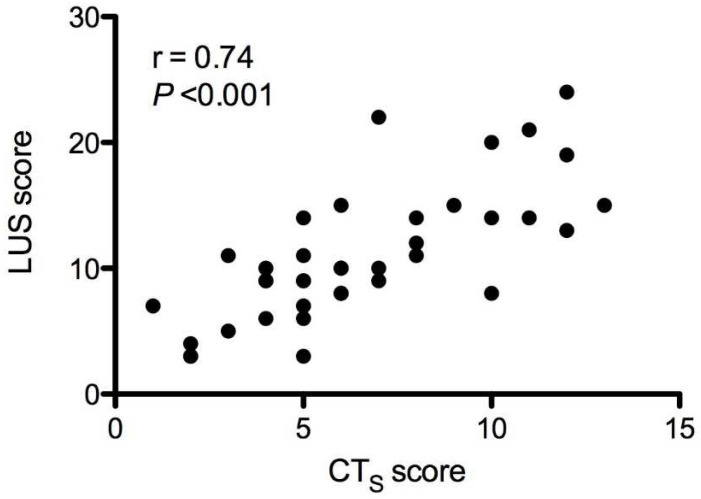
Pearson’s correlation for CTs and LUS score.

**Table 1 jcm-11-02718-t001:** Baseline demographics, clinical and laboratory data.

Variables	% or Median (IQR 25–75%)
Age, years	64 (57–72)
Gender, % male	63%
BMI, kg/m^2^	27 (25–31)
Presenting symptom, % fever or respiratory	82%
Days from onset of illness to lung ultrasound	6.5 (4–10)
ATS pneumonia severity, % severe	18%
China NHC clinical classification, % severe	76%
Pulse oximetry (SpO2), %	92 (91–93)
Supplemental Oxygen NC, L/min	2 (1–3)
White blood cell count, 10^3^/µL	5890 (4010–7645)
Neutrophil count, 10^3^/µL	4000 (2550–6100)
Lymphocyte count, 10^3^/µL	900 (600–1450)
Neutrophil-to-lymphocyte ratio	4 (3–7)
Platelet count, 10^3^/µL	191,000 (156,500–270,750)
C-reactive protein, mg/L	58 (25–106)
Lactate dehydrogenase, U/L	300 (245–442)
Haemoglobin A1c, %	6 (5.7–6.9)
D-dimer, ng/mL	673 (372–1106)
Creatinine Clearance, mL/min/1.73 m^2^	80 (63–97)

BMI, body mass index; ATS, American Thoracic Society; NHC, National Health Commission; SpO2, oxygen saturation measured by pulse oximeter; NC, nasal cannula; IQR, interquartile range.

**Table 2 jcm-11-02718-t002:** Chest computed tomography and lung ultrasound descriptive findings.

Appearance of CT Findings	Any, n (%)	Predominant, n (%)
Ground-glass opacity (±crazy paving)	36 (95%)	32 (84%)
Consolidation (±ground-glass opacity)	11 (29%)	6 (16%)
Distribution of CT findings		
Peripheral (±central)	38 (100%)	
Bilateral	36 (95%)	
Number of lobes affected, mean	4 ± 1	
1 or 2	4 (10%)	
3	7 (18%)	
4	10 (26%)	
5	17 (45%)	
Appearance of LUS findings	Any, n(%)	Predominant, n (%)
Interstitial Edema (B1 pattern)	35 (92%)	23 (61%)
Alveolar Edema (B2 pattern)	31 (82%)	16 (42%)
Consolidation (C)	11 (29%)	4 (11%)
Pleural fluid	2 (5%)	na
Distribution of LUS findings		
Bilateral	37 (97%)	
N of regions (out of 12) affected, mean	7 ± 3	

n, number per variable; na, not applicable; N, total number.

**Table 3 jcm-11-02718-t003:** Association between baseline radiographic features of lung aeration and demographic data, clinical classification, or laboratory indices of inflammation.

	Computed Tomography Software (CTs)		Lung Ultrasound(LUS)	
	Global ScoreMedian (IQR)	*p*-Value	Global ScoreMedian (IQR)	*p*-Value
Age				
<median (64 years, n = 17)	7 (5–10)	0.38	11 (8–15)	0.54
≥median (64 years, n = 21)	5 (2–8)		9 (8–14)	
Gender				
male (n = 24)	7 (4–10)	0.39	11 (8–15)	0.21
female (n = 14)	6 (4–7)		9 (6–14)	
BMI				
<median (27 kg/m^2^, n = 19)	6 (5–10)	0.65	10 (8–14)	0.99
≥median (27 kg/m^2^, n = 19)	5 (4–9)		10 (7–14)	
O2 saturation				
>median (92%, n = 17)	5 (4–6)	0.012	9 (7–11)	0.018
≤median (92%, n = 21)	8 (5–11)		14 (8–17)	
ChinaNHC classification				
moderate cases (n = 9)	4 (3–6)	0.007	9 (3–10)	0.023
severe cases (n = 29)	7 (5–10)		11 (8–15)	
ATS severity				
non-severe (n = 31)	5 (4–8)	0.045	10 (7–14)	0.14
severe (n = 7)	10 (6–12)		15 (8–20)	
Neutrophil count				
<median (4000 10^3^/µL, n = 19)	5 (4–8)	0.17	9 (6–12)	0.20
≥median (4000 10^3^/µL, n = 19)	7 (5–10)		12 (8–15)	
NLR				
<median (4, n = 19)	6 (4–9)	0.20	9 (6–14)	0.11
≥median (4, n = 19)	7 (5–11)		12 (10–14)	
C-reactive protein				
<median (58 mg/L, n = 19)	5 (3–7)	0.017	9 (5–10)	0.002
≥median (58 mg/L, n = 19)	8 (5–10)		14 (9–15)	
D-dimer				
<median (673 ng/mL, n = 19)	5 (4–7)	0.04	9 (6–11)	0.06
≥median (673 ng/mL, n = 19)	7 (5–11)		13 (8–19)	

BMI, body mass index; NHC, National Health Commission; ATS, American Thoracic Society; NLR, neutrophil-to-lymphocyte ratio.

## Data Availability

The data presented in this study are available on request from the corresponding author. The data are not publicly available due to ethical restrictions.

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
