# Peer review of "Lung Aeration in COVID-19 Pneumonia by Ultrasonography and Computed Tomography"

_jcm, 2022, doi:10.3390/jcm11102718_

Round 1
Reviewer 1 Report
This is a very good scientific study to underline the usage of PoCUS in follow-up measures to prevent radiation exposure and thus to support proof of principle.
I would suggest not to overinterprete a correlation of 0,75 and also to tell the reader the difference of CT and LUS: LUS only can detect consolidations and thus re-areation when it has attachment to the surface of the lung.
Please kindly insert in to your intro or discussion section.
Author Response
We kindly thank the reviewer for his/her remark regarding one of the most important weaknesses of lung ultrasound. There is already a short mention of this issue in the text in line 338 of the original version of the manuscript. Nevertheless, to fully address the comment of this reviewer we added the following section in lines 70-71 of the revised version as follows: ‘with main disadvantage its lower sensitivity in detecting deeper consolidations’
Reviewer 2 Report
Thank you for this interesting observational study in comparing lung ultrasound and its score to CT images for patients with non-critical Covid pneumonia. For this specific study B-lines were used to assess lung aeration when compared to GGO on CT imaging. Given bedside ultrasound is known as a valuable tool to help aid in diagnosis and management for many respiratory illnesses, it behooves clinicians to use it for all respiratory illnesses.
The design when comparing LUS to CT imaging in terms of lung aeration is reasonable, however it is already known that any interstitial process (edema, inflammation) in the lung that is seen on CT should be visible via ultrasound in terms of B-lines. Your interesting findings are more so in terms of LUS and degree of inflammation as seen by CRP levels. Given this was found, LUS could possibly be used as prognostication in terms of severity of disease and its trajectory. If the LUS has a downward trend with other date (CRP, O2 requirements, etc) then LUS become more meaningful to trend over the course of the patients hospitalization. More studies are needed for this to be conclusive however.
When comparing to CT scans I would have also repeated imaging at some point during the hospital stay to see if improvement in GGO also correlated with improvement in LUS.
I do agree that in many parts of the world a CT scanner may not be available and in that setting, clinicians should definitely be trained in bedside ultrasound (acquiring and interpreting images) for all respiratory illness. However, it should be stated that to diagnosis and or treat non-critical Covid pneumonia a CT scan of the chest is generally not required.
Overall this study is needed to further cement bedside ultrasound's role within hospital based medical practice and decrease the need for expensive and unnecessary imaging modalities.
Author Response
- The design when comparing LUS to CT imaging in terms of lung aeration is reasonable, however it is already known that any interstitial process (edema, inflammation) in the lung that is seen on CT should be visible via ultrasound in terms of B-lines. Your interesting findings are more so in terms of LUS and degree of inflammation as seen by CRP levels. Given this was found, LUS could possibly be used as prognostication in terms of severity of disease and its trajectory. If the LUS has a downward trend with other data (CRP, O2 requirements, etc) then LUS become more meaningful to trend over the course of the patients hospitalization. More studies are needed for this to be conclusive however.
ANSWER :The reviewer is right about this observation and indeed there is already some literature on this matter. However, this manuscript was mainly radiographically oriented and focused on different methods of evaluating the severity of this disease through imaging. More studies are also needed for this to be conclusive. To fully address the point of this reviewer we added the following section in lines 338-341 of the revised version as follows: ‘In addition, larger studies are needed to support the significant associations between LUS and clinical outcomes we found in our study and further promote the use of LUS as prognostication in terms of severity of COVID-19 pneumonia and its trajectory’.
- When comparing to CT scans I would have also repeated imaging at some point during the hospital stay to see if improvement in GGO also correlated with improvement in LUS.
ANSWER : We agree with the comment of the reviewer concerning the value of a repeat radiographic evaluation. This is the primary endpoint of another ongoing study during this wave of the pandemic in our institution. Nevertheless, we appreciate the comment of this reviewer and we acknowledge this limitation by adding the following section in lines 341-343 of the revised version as follows: ´In our study we do not repeat imaging at different time points during the hospital stay to examine whether improvements in GGO were also correlated with improvements in LUS´.
- I do agree that in many parts of the world a CT scanner may not be available and in that setting, clinicians should definitely be trained in bedside ultrasound (acquiring and interpreting images) for all respiratory illness. However, it should be stated that to diagnosis and or treat non-critical Covid pneumonia a CT scan of the chest is generally not required.
ANSWER : We thank the reviewer for this important comment. Following the your suggestion we added the following section in lines 48-50 of the revised version as follows: ‘Even though it is not always required for the initial diagnosis or treatment of non-critical COVID-19 infection’
Reviewer 3 Report
Kalkanis and Co-authors present a study on pulmonary aeration and lung ultrasound (LUS).
This is an interesting single center cohort aimed to compare LUS with CT scan to detect loss of ‪lung aeration in patients with a COVID-19 pneumonia
Could the Authors better define the ED where the patients were enrolled?
In April 2020 (the enrolment period) most of the affected patients were older than those reported in the present paper.
Any idea on this discrepancy on the international literature? Could it only be related to the SARS-CoV-2 storm delay in different areas?
The sample size calculation is correctly reported in the Methods section.
The supposed correlation hypothesized is quite modest (about 0.5-0.6), why the Authors chose this cut-off? They only reported a reference of the same group but several studies are available.
Did the Authors performed a sensitivity analysis on the difference sample size required for different levels of correlation?
As correctly reported by the Authors, a different lung aspect could be related to the time from the symptom onset. Could the Authors add this data?
Moreover, for the same reason, 20 hours as median interval between the LUS evaluation and the CT is a large time and differences could be related to this. Could the Authors confirm this was the first CT scan performed? Is it related to the boarding time in the Emergency Department?
Interestingly, the Authors reported B-lines in 37 patients and ‪ground-glass opacity in 36 at the CT scan. How they explain this result? Could some patients have suffered from COVID19-related pneumonia and any form of congestion? In this case, the time between the evaluation could have a relevant burden?
How is the Bland-Altman plot for LUS and CTs score?
The Authors measured the loss of aeration with an ad hoc software. Why they decided to convert it in a scale instead of converting the LUS score in a percentage of loss of aeration? It could be more interesting.
What the clinical meaning that the Authors refer to the present result?
Diagnostic o prognostic? In both cases, some additional analyses seem to be needed.
Some references seem to be missing (e.g., ‪International Consensus Conference on lung ultrasound is reported but without the appropriate reference)
Author Response
- Could the Authors better define the ED where the patients were enrolled?
ANSWER : A clarifying sentence regarding the ED of our tertiary university center was added in the text in lines 113-115 of the revised version that reads as follows: ´This was part of the emergency department (ED) planning of our tertiary university hospital, in order to immediately isolate patients with flu-like symptoms while all the needed tests were performed`.
- In April 2020 (the enrolment period) most of the affected patients were older than those reported in the present paper. Any idea on this discrepancy on the international literature? Could it only be related to the SARS-CoV-2 storm delay in different areas?
ANSWER : The most logical explanation for this discrepancy is that in Belgium the great majority of the older patients during the course of the study where immediately hospitalized to the medium care or the intensive care units and not in emergency department, thus excluding them from the enrollment in this study.
- The sample size calculation is correctly reported in the Methods section. The supposed correlation hypothesized is quite modest (about 0.5-0.6), why the Authors chose this cut-off? They only reported a reference of the same group but several studies are available. Did the Authors performed a sensitivity analysis on the difference sample size required for different levels of correlation?
ANSWER : We thank the reviewer for allowing us to clarify this concern. As the present study constitutes a part of a larger project that is conducted in patients hospitalized for COVID-19 pneumonia in Belgium, we preferred to use this cut-off, produced in the same group of Belgian patients- and indicated significant associations between LUS score and other clinical outcomes (correlation r ranged between 0.48-0.58). No, we did not perform a sensitivity analysis on the different sample sizes required for different levels of correlation. We appreciate the comment of this reviewer and we now address his/her important comment in lines 167-170 of the revised version as follows: ´We did not perform a sensitivity analysis on the different sample sizes required for different levels of correlation between LUS and CT scores to determine the optimal sample size of this study to detect statistical significance´.
- As correctly reported by the Authors, a different lung aspect could be related to the time from the symptom onset. Could the Authors add this data?
ANSWER : This is a very interesting point from the reviewer. Unfortunately, because of a) great discrepancy of the stating of the symptoms through the patients, or the GPs, during this first wave of the pandemic, and b) the lack of a follow-up imaging in the methodology of this study, we decided not to include this data. Other larger studies have showed this dynamic evolution of the CT imaging in the acute phase of the infection and in patient groups of different severity. Please see the reference below.
Wang YC, Luo H, Liu S, et al. Dynamic evolution of COVID-19 on chest computed tomography: experience from Jiangsu Province of China. Eur Radiol. 2020;30(11):6194-6203. doi:10.1007/s00330-020-06976-6
- Moreover, for the same reason, 20 hours as median interval between the LUS evaluation and the CT is a large time and differences could be related to this. Could the Authors confirm this was the first CT scan performed? Is it related to the boarding time in the Emergency Department? Interestingly, the Authors reported B-lines in 37 patients and ‪ground-glass opacity in 36 at the CT scan. How they explain this result? Could some patients have suffered from COVID19-related pneumonia and any form of congestion? In this case, the time between the evaluation could have a relevant burden?
ANSWER : This is another important observation raised by this reviewer. We also thank the reviewer for giving us the opportunity to clarify this issue. Indeed, this was the first CT performed to these patients. The unfortunate but unavoidable time delay between the two imaging procedures is related to the structure of the whole system of managing COVID patients in the ED department and to the priority of transferring more seriously ill patients first to the CT scanner and the intensive / medium care wards. As correctly noted by the reviewer the median age of our cohort was relatively low and no patient suffered from serious heart failure. To our knowledge, there is no ultrasonographic study with an intermediate period between two consecutive measurements shorter than 48 hours. We considered 24 hours a reasonable period to ensure the optimal care of the patient, the uninterrupted flow of the ED and the safety of echographer.
Hoffmann T, Bulla P, Jödicke L, Klein C, Bott SM, Keller R, Malek N, Fröhlich E, Göpel S, Blumenstock G, Fusco S. Can follow up lung ultrasound in Coronavirus Disease-19 patients indicate clinical outcome? PLoS One. 2021 Aug 25;16(8):e0256359. doi: 10.1371/journal.pone.0256359. PMID: 34432835; PMCID: PMC8386874.
- How is the Bland-Altman plot for LUS and CTs score?
ANSWER : We argue that a Bland-Altman plot for LUS and CTs score is not meaningful to be presented since LUS and CTs scoring system is different to allow a comparison between the two methods.
- The Authors measured the loss of aeration with an ad hoc software. Why they decided to convert it in a scale instead of converting the LUS score in a percentage of loss of aeration? It could be more interesting.
ANSWER : The lung ultrasound score (LUS) is a semi quantitative score that measures lung aeration loss Converting the LUS score in a percentage of loss of aeration has not been in the same extent validated in pneumonia like the LUS score. Certainly, the proposal of this reviewer constitutes a very interesting idea for future research in this topic of interest.
- What the clinical meaning that the Authors refer to the present result? Diagnostic o prognostic? In both cases, some additional analyses seem to be needed.
ANSWER : This is an observational study to primarily examine the diagnostic value of lung ultrasound in COVID-19 pneumonia. However, as reviewers 1 and 2 also highlighted in their report, we acknowledge that a) we do not repeat imaging at different time points during the hospital stay for examining whether improvements in GGO were also correlated with improvements in LUS and b) that larger studies are needed to support the significant associations between LUS and clinical outcomes to promote the use of LUS as prognostication in terms of severity of COVID-19 pneumonia and its trajectory (please see in lines 338-341 of the revised version).
- Some references seem to be missing (e.g., ‪International Consensus Conference on lung ultrasound is reported but without the appropriate reference)
ANSWER : Thank you for spotting this issue.
Round 2
Reviewer 3 Report
I would like to thank the Authors for the time they used for answering my doubts.
The Authors seems to state that the study was partially (?) performed in the Emergency Department (as added in order to answer one of my question) but the they also reported “older patients during the course of the study where immediately hospitalized to the medium care or the intensive care units and not in emergency department” so there likely was a selection bias in the included patients due to the logistics of the hospital where the study was held?
I am sorry for this comment but the answer made it.
I understand the answer about timing but, due to the lung ultrasound characteristics, information about symptoms onset seem to be crucial to me.
It seems odd that the Authors reported “no ultrasonographic study with an intermediate period between two consecutive measurements shorter than 48 hours”, did they refer to the COVID19?
In the ED, several studies were performed with assessment of LUS and another measurement among undifferentiated dyspneas (e.g., studies of different Italian groups in Florence and Turin, or from the MGH group in Boston) as well as including COVID19 pneumonias.
Twenty- four hours are definitively a reasonable time in a clinical prospective but not in for using LUS and another measurement based on the sonographic intrinsic characteristics.
Author Response
Comments and Suggestions for Authors:
I would like to thank the Authors for the time they used for answering my doubts.
Our team would like to thank the reviewer for his interest and his insightfull comments
1. The Authors seems to state that the study was partially (?) performed in the Emergency Department (as added in order to answer one of my question) but the they also reported “older patients during the course of the study where immediately hospitalized to the medium care or the intensive care units and not in emergency department” so there likely was a selection bias in the included patients due to the logistics of the hospital where the study was held?
I am sorry for this comment but the answer made it.
We thank the reviewer for giving us the chance to clarify this matter. As stated in the design, the study was performed in the non-critical care ward, were the LUS took place. Thus, patients that were directly admitted to critical care were de facto excluded. The CT was dedicated to the ED and the scanning was performed at the time of admission to all COVID positive patients according to hospital policy at that time.
We believe there was no selection bias, as this study, per design, targeted already admitted patients in the ward. We have added this sentence to the discussion, line 332: "We can’t generalize our results to patients admitted immediately to critical care."
2. I understand the answer about timing but, due to the lung ultrasound characteristics, information about symptoms onset seem to be crucial to me.
The reviewer is right about the importance of time between time-onset and ultrasonographic examination. This data was extracted from our database and added to the manuscript in Table 1.
3. It seems odd that the Authors reported “no ultrasonographic study with an intermediate period between two consecutive measurements shorter than 48 hours”, did they refer to the COVID19?
In the ED, several studies were performed with assessment of LUS and another measurement among undifferentiated dyspneas (e.g., studies of different Italian groups in Florence and Turin, or from the MGH group in Boston) as well as including COVID19 pneumonias.
Twenty- four hours are definitively a reasonable time in a clinical prospective but not in for using LUS and another measurement based on the sonographic intrinsic characteristics.
We thank the reviewer for giving us the chance to clarify this. We commented in our answer about the lack of literature regarding two consecutive ultrasonographic measurements in less than 48 hours. Based on the existing knowledge and the experience of our department regarding the chronological evolution of the radiographic presentation of COVID disease we considered 24h a reasonable period to add as a criterium while designing our protocol.
In reality the maximum time from admission to the LUS was 12 hours and it was added to the text, line 139. We understand that even this delay makes the correlation between the CT and LUS scores suboptimal and it was added as a weak point of the study: line 326